# QXplore: Q-learning Exploration by Maximizing Temporal Difference Error

## Abstract

A major challenge in reinforcement learning is *exploration*, especially when reward landscapes are sparse. Several recent methods provide an intrinsic motivation to explore by directly encouraging agents to seek novel states. A potential disadvantage of pure state novelty-seeking behavior is that unknown states are treated equally regardless of their potential for future reward. In this paper, we propose an exploration objective using the temporal difference error experienced on extrinsic rewards as a secondary reward signal for exploration in deep reinforcement learning. Our objective yields novelty-seeking in the absence of extrinsic reward, while accelerating exploration of reward-relevant states in sparse (but nonzero) reward landscapes. We implement the objective with a two-policy Q-learning method in which $Q$ and $Q_x$ are the action-value functions for extrinsic and secondary rewards, respectively. Secondary reward is given by the absolute value of the TD-error of $Q$. Training is off-policy, based on a replay buffer containing a mix of trajectories sampled using $Q$ and $Q_x$. We characterize performance on a set of continuous control benchmark tasks, and demonstrate comparable or faster convergence on all tasks when compared with other state-of-the-art exploration methods.

## 1 Introduction

Deep reinforcement learning (RL) has recently achieved impressive results across several challenging domains, such as playing games (Mnih et al., 2016; Silver et al., 2017; OpenAI, 2018; Baker et al., 2019) and controlling robots (OpenAI et al., 2018; Kalashnikov et al., 2018). In many of these tasks, a well-shaped reward function is critical to learning performant policies. On the other hand, deep RL still remains challenging for tasks where the reward function is sparse. In these settings, state-of-the-art RL methods often perform poorly and train very slowly, if at all, due to the low probability of observing improved rewards by following the current optimal policy or with a naive exploration policy such as $\epsilon$-greedy sampling.

The challenge of learning from sparse rewards is typically framed as a problem of *exploration*, inspired by the notion that a successful RL agent must efficiently explore the state space of its environment in order to find improved sources of reward. One common exploration paradigm is to directly determine the novelty of states and to encourage the agent to visit states with the highest novelty. In small MDPs this can be achieved through counting how many times each state has been visited. This approach often performs poorly in high-dimensional or continuous state spaces, but recent work (Tang et al., 2017; Bellemare et al., 2016; Fu et al., 2017) using count-like statistics have shown success on benchmark tasks with complex state spaces. Another paradigm for exploration learns a dynamic model of the environment and computes a novelty measure proportional to the error of the model in predicting transitions in the environment. This exploration method relies on the core assumption that well-modeled regions of the state space are similar to previously visited states and thus are less interesting than other regions of state space. Predictions of the transition dynamics can be directly computed (Pathak et al., 2017; Stadie et al., 2015; Savinov et al., 2019; Burda et al., 2019a), or related to an information gain objective on the state space, as described in VIME (Houthooft et al., 2016) and EMI (Kim et al., 2018).

Several exploration methods have recently been proposed that capitalize on the function approximation properties of neural networks. Random network distillation (RND) trains a function to predict the output of a randomly-initialized neural network from an input state, and uses the approximation

error as a reward bonus for a separately-trained RL agent (Burda et al., 2019b). Similarly, DORA (Fox et al., 2018) trains a network to predict zero on observed states and deviations from zero are used to indicate unexplored states.

An important shortcoming of existing exploration methods is that they only incorporate information about states and therefore assume all unobserved states are equally motivating, regardless of their viability for future reward. The viability of this assumption is highly task dependent: While games like Montezuma's Revenge or Super Mario Bros, where novelty correlates highly with success, can be attacked effectively by state novelty methods alone (Burda et al., 2019b; Pathak et al., 2017; Ecoffet et al., 2019; Kim et al., 2018), other tasks such as hide-and-seek or some Atari games where novelty and utility are less correlated tend to frustrate state novelty methods (Burda et al., 2019b; Baker et al., 2019; Burda et al., 2019a). Baker et al. (2019) explored using both RND and a simple state counting baseline to discover skills such as navigation and block-pushing in a hide-and-seek environment. However, the authors found that careful construction of the state representation used for novelty seeking was necessary to discover any such skills, as novelty in the full state space did not correspond to novelty in the intuitive sense (Baker et al., 2019).

Instead of focusing on the state-space, this work uses the temporal difference error (TD-error) which provides a signal into novelty in the reward landscape. Past works have also utilized information from the reward landscape as a learning signal to various extents. Schmidhuber et. al. first describe using reward misprediction and model prediction error for exploration (Schmidhuber, 1991; Thrun & Möller, 1991; 1992). However, the work was primarily concerned with model-building and system-identification in small MDPs, and used reward prediction error rather than TD-error. Later, Gehring & Precup (2013) used TD-error as a negative signal to constrain exploration to focus on states that are well understood by the value function to avoid common failure modes. Related to maximizing TD-error is maximizing the variance or KL-divergence of a posterior distribution over MDPs or Q-functions, which can be used as a measure of uncertainty (Osband & Van Roy, 2017; O'Donoghue et al., 2017; Chen et al., 2017; Fox et al., 2018; Osband et al., 2018). Posterior uncertainty over Q-functions can be used for information gain in the reward or Q-function space, as opposed to information gain in the state space as described by VIME among others (Houthooft et al., 2016; Kim et al., 2018), though posterior uncertainty methods have thus-far largely been used for local exploration as an alternative to dithering methods such as $\epsilon$-greedy sampling, though Osband et al. (2018) do apply posterior uncertainty to Montezuma's Revenge.

In this paper we propose QXplore, a new exploration formulation that seeks novelty in the predicted reward landscape instead of novelty in the state space. QXplore exploits the inherent reward-space signal from the computation of temporal difference error (TD-error) in value-based RL, and explicitly promotes visiting states where the current understanding of reward dynamics is poor. In the following sections, we describe QXplore and demonstrate its utility for efficient learning on a variety of complex benchmark environments with continuous controls and sparse rewards.

## 2 PRELIMINARIES

We consider RL in the terminology of Sutton & Barto (1998), in which an agent seeks to maximize reward in a Markov Decision Process (MDP). An MDP consists of states $s \in \mathcal{S}$, actions $a \in \mathcal{A}$, a state transition function $S : \mathcal{S} \times \mathcal{A} \times \mathcal{S} \rightarrow [0, 1]$ giving the probability of moving to state $s_{t+1}$ after taking action $a_t$ from state $s_t$ for discrete timesteps $t \in 0, ..., T$. Rewards are sampled from reward function $r : \mathcal{S} \times \mathcal{A} \rightarrow \mathbb{R}$. An RL agent has a policy $\pi(s_t, a_t) = p(a_t|s_t)$ that gives the probability of taking action $a_t$ when in state $s_t$. The agent aims to learn a policy to maximize the expectation of the time-decayed sum of reward $R_\pi(s_0) = \sum_{t=0}^{T} \gamma^t r(s_t, a_t)$ where $a_t \sim \pi(s_t, a_t)$.

A value function $V_\theta(s_t)$ with parameters $\theta$ is a function which computes $V_\theta(s_t) \approx R_\pi(s_t)$ for some policy $\pi$. Temporal Difference (TD) error $\delta_t$ measures the bootstrapped error between the value function at the current timestep and the next timestep as

$$\delta_t = V_\theta(s_t) - (r(s_t, a_t \sim \pi(s_t)) + \gamma V_\theta(s_{t+1})). \tag{1}$$

A Q-function is a value function of the form $Q(s_t, a_t)$, which computes $Q(s_t, a_t) = r(s_t, a_t) + \gamma \cdot \max_{a'} Q(s_{t+1}, a')$, the expected future reward assuming the optimal action is taken at each future timestep. An approximation to this optimal Q-function $Q_\theta$ with some parameters $\theta$ may be trained using a mean squared TD-error objective $L_{Q_\theta} = ||Q_\theta(s_t, a_t) - (r(s_t, a_t) + \gamma \cdot \max_{a'} Q'_{\theta'}(s_{t+1}, a'))||^2$

given some target Q-function $Q'_{\theta'}$, commonly a time-delayed version of $Q_\theta$ (Mnih et al., 2015). Extracting a policy $\pi$ given $Q_\theta$ amounts to approximating $\text{argmax}_a Q_\theta(s_t, a)$. Many methods exist for approximating the $\text{argmax}_a$ operation in both discrete and continuous action spaces (Lillicrap et al., 2015; Haarnoja et al., 2018). Following the convention of Mnih et al. (2016), we train $Q_\theta$ using an off-policy replay buffer of previously visited $(s, a, r, s')$ tuples, which we sample uniformly.

## 3  QXPLORE: TD-ERROR AS ADVERSARIAL REWARD SIGNAL

### 3.1  METHOD OVERVIEW

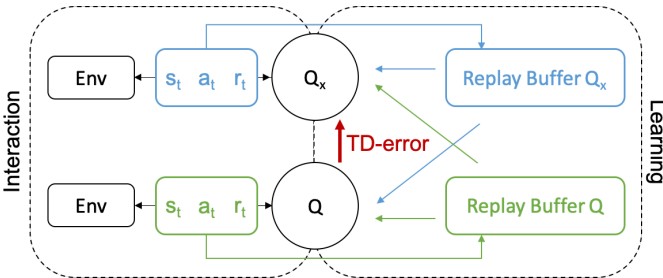

Figure 1: Method diagram for QXplore. We define two Q-functions which sample trajectories from their environment and store experiences in separate replay buffers. $Q$ is a standard state-action value-function, whereas $Q_x$'s reward function is the unsigned temporal difference error of the current Q on data sampled from both replay buffers. A policy defined by $Q_x$ samples experiences that maximize the TD-error of $Q$, while a policy defined by $Q$ samples experiences that maximize discounted reward from the environment.

We first provide an overview of the method - a visual representation is depicted in Figure 1. At a high level, QXplore is an exploration method that jointly trains two independent agents equipped with their own Q-functions and reward functions:

1. $Q$: A standard Q-function, that learns a value function on reward provided by the external environment.

2. $Q_x$: A Q-function that learns a value function directly on the TD-error of $Q$.

The policy $\pi_{Q_x}$ that samples $Q_x$ and the Q-function $Q$ form an adversarial pair, wherein $\pi_{Q_x}$ seeks to sample state-action pairs that produce large TD-errors while $Q$'s training objective $L_{Q_\theta}$ attempts to minimize the TD-error for previously sampled state-action pairs. Thus, $\pi_{Q_x}$ achieves reward when the agent ventures into states whose reward dynamics are foreign to $Q$ (i.e. $Q$ under/overestimates reward achieved). Separate replay buffers are maintained for each agent, but each agent receives samples from both buffers at train time. A similar adversarial sampling scheme was used to train an inverse dynamics model by Hong et al. (2018), and Colas et al. (2018) use separate goal-driven exploration and reward maximization phases for efficient learning, but to our knowledge parallel adversarial sampling policies have not previously been used for exploration.

### 3.2  TD-ERROR OBJECTIVE

We directly treat TD-error as a reward signal and use a Q-function trained on this signal to induce an exploration policy, rather than as a supplementary objective or to compute a confidence bound. Crucially, when combined with neural network function approximators, this signal provides meaningful exploration information everywhere as discussed in Section 3.4. For a value function with parameters $\theta$, and TD-error $\delta_t$ we define our exploration reward function as

$$r_{x,\theta}(s_t, a_t, s_{t+1}) = |\delta_t| = |Q_\theta(s_t, a_t) - (r_\text{E}(s_t, a_t) + \gamma\max_{a'} Q'_{\theta'}(s_{t+1}, a'))| \qquad (2)$$

for some extrinsic reward function $r_\text{E}$ and target Q-function $Q'_\theta$. Notably, we use the absolute value of the temporal difference (rather than the squared error) used to compute updates for $Q_\theta$ to keep the magnitudes of $r_\text{E}$ and $r_x$ comparable and reduce the influence of outlier temporal differences on the gradients of $Q_x$, which we describe below.

---

**Algorithm 1** QXplore Algorithm

---

**Input:** MDP $S$, Q-function $Q_\theta$ with target $Q'_{\theta'}$, $Q_x$ function $Q_{x,\phi}$ with target $Q'_{x,\phi'}$, replay buffers $\mathscr{Z}_Q$ and $\mathscr{Z}_{Q_x}$, batch size $B$ and sampling ratios $\mathcal{R}_Q$ and $\mathcal{R}_{Q_x}$, CEM policies $\pi_Q$ and $\pi_{Q_x}$, time decay parameter $\gamma$, soft target update rate $\tau$, and environments $E_Q, E_{Q_x}$

**while** not converged **do**

    Reset $E_Q, E_{Q_x}$

    **while** $E_Q$ and $E_{Q_x}$ are not done **do**

        **Sample environments**

        $\mathscr{Z}_Q \leftarrow (s, a, r, s') \sim \pi_Q | E_Q$

        $\mathscr{Z}_{Q_x} \leftarrow (s, a, r, s') \sim \pi_{Q_x} | E_{Q_x}$

        **Sample minibatches for $Q_\theta$ and $Q_{x,\phi}$**

        $(s_Q, a_Q, r_Q, s'_Q) \leftarrow B * \mathcal{R}_Q$ samples from $\mathscr{Z}_Q$ and $B * (1 - \mathcal{R}_Q)$ samples from $\mathscr{Z}_{Q_x}$

        $(s_{Q_x}, a_{Q_x}, r_{Q_x}, s'_{Q_x}) \leftarrow B * \mathcal{R}_{Q_x}$ samples from $\mathscr{Z}_{Q_x}$ and $B * (1 - \mathcal{R}_{Q_x})$ samples from $\mathscr{Z}_Q$

        **Train**

        $r_{x,\theta} \leftarrow |Q_\theta(s_{Q_x}, a_{Q_x}) - (r_{Q_x} + \gamma Q'_{\theta'}(s'_{Q_x}, \pi_Q(s'_{Q_x})))|$

        $L_Q \leftarrow ||Q_\theta(s_Q, a_Q) - (r_Q + \gamma Q'_{\theta'}(s'_Q, \pi_Q(s'_Q)))||^2$

        $L_{Q_x} \leftarrow ||Q_{x,\phi}(s_{Q_x}, a_{Q_x}) - (r_{x,\theta} + \gamma Q'_{x,\phi'}(s'_{Q_x}, \pi_{Q_x}(s'_{Q_x})))||^2$

        Update $\theta \propto L_Q$

        Update $\phi \propto L_{Q_x}$

        $\theta' \leftarrow (1 - \tau)\theta' + \tau\theta$

        $\phi' \leftarrow (1 - \tau)\phi' + \tau\phi$

    **end while**

**end while**

---

Intuitively, a policy maximizing the expected sum of $r_x$ will sample trajectories where $Q_\theta$ does not have an accurate estimate of the future rewards it will experience. This is useful for exploration because $r_x$ will be large not only for state-action pairs producing unexpected reward, but for all state-action pairs leading to such states, providing a much denser exploration reward function. Further, TD-error-based exploration with a dedicated exploration policy removes the exploration-versus-exploitation tradeoff that state-novelty methods must contend with, where trajectories maximizing state novelty often do not also maximize reward. Separate exploration and exploitation policies allow us to sample trajectories maximizing $r_x$ that provide information about the task for $Q_\theta$ to train on without impacting its ability to maximize reward.

### 3.3   $Q_x$: LEARNING A Q-FUNCTION TO MAXIMIZE TD-ERROR

Next, we will describe how we use the TD-error signal defined in Section 3.2 to define an exploration policy. The reward function $r_x$ is generic, and can be maximized by any RL algorithm. However, given its derivation from a bootstrapped Q-function, training a second Q-function to maximize $r_x$ allows the entire algorithm to be trained off-policy with two replay buffers that share data between $Q_\theta$ and the Q-function maximizing $r_x$, which we term $Q_x$. This approach is beneficial for exploration, as it avoids needing to trade off between exploration and exploitation via a weighting hyperparameter, and sharing data between replay buffers improves data efficiency for training both Q-functions.

We define a Q-function, $Q_{x,\phi}(s, a)$ with parameters $\phi$, whose reward objective is $r_x$. We train $Q_{x,\phi}$ using the standard bootstrapped loss function

$$L_{Q_{x,\phi}} = ||Q_{x,\phi}(s_t, a_t) - (r_x(s_t, a_t, s_{t+1}) + \gamma \max_{a'} Q'_{x,\phi'}(s_{t+1}, a'))||^2. \tag{3}$$

The two Q-functions, $Q_\theta$ and $Q_x$, are trained off-policy in parallel, sharing replay data so that $Q_\theta$ can train on sources of reward discovered by $Q_x$ and so that $Q_x$ can better predict the TD-errors of $Q_\theta$. Since the two share data, $\pi_{Q_x}$ acts as an adversarial teacher for $Q_\theta$, sampling trajectories that produce high TD-error under $Q_\theta$ and thus provide novel information about the reward landscape. To avoid off-policy stability issues due to the different reward objectives, we sample a fixed ratio of experiences collected by each policy for each training batch. We use a nonparametric cross-entropy method policy inspired by Kalashnikov et al. (2018), previously described as more robust to hyperparameter variance (Simmons-Edler et al., 2019; Kalashnikov et al., 2018). We also experimented with a variant using

DDPG-style parametric policies (Lillicrap et al., 2015) for both $Q_\theta$ and $Q_x$, but found preventing sampling collapse by $Q_\theta$'s policy difficult. Our full method is shown in Figure 1, and pseudocode in Algorithm 1.

### 3.4 STATE NOVELTY FROM NEURAL NETWORK FUNCTION APPROXIMATION ERROR

A key question in using TD-error for exploration is: What happens when the reward landscape is flat? Theoretically, in the case that $\forall(s, a), r(s, a) = C$ for some constant $C \in \mathbb{R}$, an optimal Q-function which generalizes perfectly to unseen states will, in the infinite time horizon case, simply output $\forall(s, a), Q^\star(s, a) = \sum_{t=0}^{\infty} C\gamma^t$. This results in a TD-error of 0 everywhere and thus no exploration signal. However, using neural network function approximation, we find that perfect generalization to unseen states-action pairs does not occur, and in fact observe in Figure 2 that the distance of a new datapoint from the training data manifold correlates with the magnitude of the network output's deviation from $\sum_{t=1}^{\infty} C\gamma^t$ and thus with TD-error. As a result, in the case where the reward landscape is flat TD-error exploration converges to a form of state novelty exploration. This property of neural network function approximation has been used by several previous exploration methods to good effect, including RND (Burda et al., 2019b) and DORA (Fox et al., 2018). In particular, the exploration signal used by RND (extrapolation error from fitting the output of a random network) should be analogous to $r_x$ (extrapolation error from fitting a constant value), meaning we should expect to perform comparably to RND in the worst case where no extrinsic reward exists.

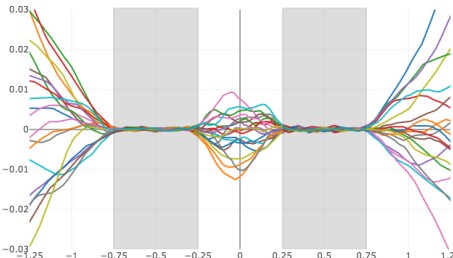

Figure 2: A neural network trained to predict a constant value does not interpolate or extrapolate well outside its training range, which can be exploited for exploration. Predictions of 3-layer MLPs of 256 hidden units per layer trained to imitate $f(x) = 0$ on $\mathbb{R} \to \mathbb{R}$ with training data sampled uniformly from the range $[-0.75, -0.25] \cup [0.25, 0.75]$. Each line is the final response curve of an independently trained network once its training error has converged (MSE < 1e-7).

## 4 EXPERIMENTS

We performed several experiments to demonstrate the effectiveness of $Q_x$ on continuous control benchmark tasks. We compare QXplore with a related state of the art state novelty-based method, RND (Burda et al., 2019b), DORA (Fox et al., 2018), and with $\epsilon$-greedy sampling as a simple baseline. Each method is implemented in a shared code base on top of TD3 Fujimoto et al. (2018b) using a cross entropy method policy as proposed by Qt-Opt Kalashnikov et al. (2018) for hyperparameter stability. We also compare to results from several previous works on `SparseHalfCheetah`. Finally, we present several ablations to QXplore, as well as analysis of its robustness in response to several hyperparameters. Implementation details and hyperparameters for QXplore, RND, DORA, and $\epsilon$-greedy can be found in Appendix A.

### 4.1 EXPERIMENTAL SETUP

We benchmark on four continuous control tasks using the MuJoCo physics simulator that each require exploration due to sparse rewards. First, the `SparseHalfCheetah` task originally proposed by VIME (Houthooft et al., 2016). Next, we benchmark on three OpenAI gym tasks, `FetchPush`, `FetchSlide` and `FetchPickAndPlace`, originally developed for goal-directed exploration methods such as HER (Andrychowicz et al., 2017). We chose these tasks as they are challenging exploration problems that are relatively simple to control, but still involve large continuous state spaces and in the case of the `Fetch` tasks learning to generalize across random object/goal positions. For consistent reward shaping across tasks we used a reward function in the range [-1-0] for

`SparseHalfCheetah` similar to the `Fetch` tasks, but results on the original reward function from Houthooft et al. (2016) can be found in Appendix E, where we perform comparably. We ran 5 random seeds for each experiment. More details on these environments can be found in Appendix B.

## 4.2 EXPLORATION BENCHMARK PERFORMANCE

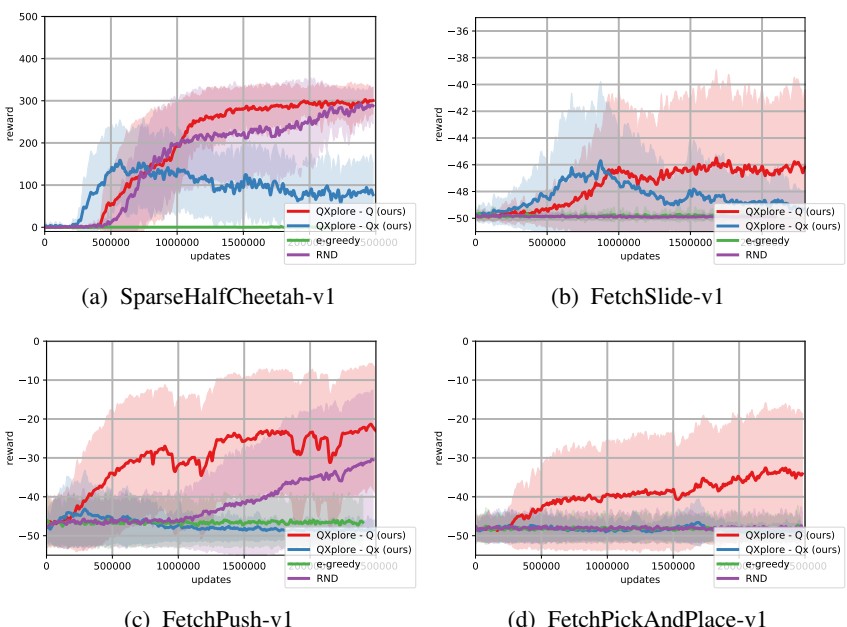

(a) SparseHalfCheetah-v1      (b) FetchSlide-v1

(c) FetchPush-v1      (d) FetchPickAndPlace-v1

Figure 3: Performance of QXplore compared with RND and $\epsilon$-greedy sampling. QXplore outperforms RND and $\epsilon$-greedy on continuous control tasks. QXplore performs better due to efficient exploration sampling by $Q_x$ and the separation of the exploration and exploitation objectives. $Q$ indicates the performance of our exploitation Q-function, while $Q_x$ indicates the performance of our exploration Q-function, whose objective does not directly maximize reward but which may lead to high reward regardless.

| Episodes until mean reward of | QXplore | VIME | EX2 | EMI | GEP-PG | DORA | SimHash |
|---|---|---|---|---|---|---|---|
| 50 | 2000 | 10000* | 4740* | 2580* | NA | x | x* |
| 100 | 3000 | x* | 6180* | 4520* | 4000 | x | x* |
| 200 | 4000 | x* | x* | 8440* | x | x | x* |
| 300 | 7900 | x* | x* | x* | x | x | x* |

Table 1: Number of episodes required to reach mean reward milestones on `SparseHalfCheetah` for several methods. QXplore outperforms previously published methods. Results marked with "*" are previously published numbers. VIME from Houthooft et al. (2016), EMI and EX2 from Kim et al. (2018), and SimHash from Tang et al. (2017). Results marked with "x" indicate that the mean reward was not achieved. For GEP-PG (Colas et al., 2018) we used the author's implementation, which did not permit easy evaluation of intermediate performance.

We show the performance of each method on each task in Figure 3. QXplore outperforms RND modestly on the `SparseHalfCheetah` task, but performs much better comparatively on the `Fetch` tasks- only on `FetchPush`, the easiest task, did RND find non-random reward. We theorize that this improved performance on the `Fetch` tasks is because QXplore's TD-error exploration drives the agent to discover the conditional relationship between the changing goal position and the reward function, whereas RND and other state novelty methods are goal-agnostic since the goal is static for the entire episode. While QXplore is not a goal-directed RL method, and does not achieve state-of-the-art performance compared to dedicated goal-directed RL methods, the fact that this relationship is discovered through TD-error exploration is encouraging as to its broader applicability.

We compare to several other exploration methods is in Table 1. The methods from previous work are built on top of TRPO (Schulman et al., 2015), so a comparison in terms of training iterations as in Figure 3 would not be informative due to TRPO's variable update rule. We instead compare the number of episodes of interaction required to reach a given level of reward, though QXplore was not intended to be performant with respect to this metric. While some decrease in episode efficiency is expected due to differing baseline methods (TRPO (Schulman et al., 2015) versus TD3 Fujimoto et al. (2018b)), compared to published results for EMI (Kim et al., 2018), EX2 (Fu et al., 2017), VIME (Houthooft et al., 2016), and SimHash (Tang et al., 2017) on the `SparseHalfCheetah` task, QXplore reaches every reward milestone faster, and achieves a peak reward (300) not achieved by any previous method.

We also include here the performance of our implementation of DORA (Fox et al., 2018) on `SparseHalfCheetah`. DORA performed poorly, possibly because it was not intended for use with continuous action spaces, and thus we did not test it on other tasks.

Finally, we compare to GEP-PG (Colas et al., 2018), which used separate exploration and exploitation phases similar to QXplore. We downloaded the author's implementation (built on top of DDPG) and tested it on `SparseHalfCheetah` using the parameters for the `HalfCheetah-v2` task it was originally tested on. The author's implementation did not facilitate evaluating performance midway through training, and thus we report only their final performance number after 4000 episodes, which was 120.2.

## 4.3 ROBUSTNESS

As RL tasks are highly heterogeneous, and good parameterization/performance can be hard to obtain in practice for many methods (Henderson et al., 2018), we performed sweeps over several hyperparameters and introduce several ablations of QXplore on `SparseHalfCheetah` to demonstrate the method's robustness and validate aspects of the algorithm.

**Parameter Sweeps** We swept over the learning rates of $Q$ and $Q_x$, as well as the ratio of self-collected versus other-collected data used to train each function. The results suggest that while the performance of $Q$ is somewhat sensitive to learning rate, keeping learning rates for $Q$ and $Q_x$ the same works well. The results also show that while our ratio of 75% self/25% non-self performs best, $Q$ is fairly robust to the on/off-policy data ratio, including when $Q$ is trained entirely off-policy on data collected by $Q_x$. Results are shown in Figures 8 and 9 in Appendix D.

**Weight Initialization** Also, since neural network generalization is key to QXplore, we tested several different network weight initialization schemes, including some that were deliberately poor priors. We found that while the performance of $Q$ is sensitive to initialization scheme, $Q_x$ robustly finds reward in all cases. See Figure 12 in Appendix G.

**The 'Noisy TV' Problem** One drawback that naive state novelty exploration methods have is that unpredictable observations (such as from a TV displaying static) act as maxima in the exploration reward function. Naive methods are unavoidably drawn to such states instead of exploring. TD-error driven exploration is not sensitive to unpredictable observations as they do not affect the underlying reward function. To demonstrate this, we tested QXplore with a variant of the `SparseHalfCheetah` task with noisy observations. We observe that QXplore performs as normal in this case. A description of the task can be found in Appendix F.

## 4.4 ABLATIONS

There are two features of QXplore that distinguish it from prior work in exploration: the use of a pair of policies that share replay data, the use of unsigned TD-Error to drive exploration. We performed several ablations that assess the impact of aspects of each of these features. Detailed results can be found in Appendix C.

**Single-Policy QXplore** First, we test a single-policy version of QXplore by replacing $Q_\theta(s, a)$ with a value function $V_\theta(s)$. We use a value function rather than Q-function in this case to avoid large estimation errors stemming from fully off-policy training such as reported by Fujimoto et al. (2018a). We observe in Figure 5 that while the policy is able to find reward quickly and converge faster, the need to satisfy both objectives results in a lower converged reward than the original QXplore method.

**1-Step Reward Prediction** Second, we run an ablation where we replaced $Q_\theta(s, a)$ with a function that simply predicts the current $r(s_t, a_t)$. Using reward error instead of a value function in $Q_x$ can still produce the same state novelty fallback behavior in the absence of reward; however, it provides only limited reward-based exploration utility. We tested this variant and observe in Figure 5 that it fails to sample reward. Reward prediction error is not sufficient to allow strong exploration behavior.

**QXplore with State Novelty Exploration** To assess the importance of TD-error specifically in our two policy algorithm, we replaced the TD-error maximization objective of $Q_x$ with the random network prediction error maximization objective of RND, while still performing separate rollouts of each policy. The results are shown in Figure 6. We observe that while the modified $Q_x$ function does sample reward, it is too infrequent to guide $Q$ to learn the task, and further that the modified $Q_x$ function does not display directional preference in exploration once reward is discovered.

**QXplore with Signed TD-Error Objective** While we used unsigned TD-error to train $Q_x$, we also tested QXplore using signed TD-error. We used the negative signed TD-error $-\delta_t$ from equation 1 so that better-than-expected rewards result in positive $r_x$ values. The results of this experiment are shown in Figure 7. The unsigned TD-error performs better on `SparseHalfCheetah`.

### 4.5 QUALITATIVE BEHAVIORAL ANALYSIS

Qualitatively, on `SparseHalfCheetah` we observe interesting behavior from $Q_x$ late in training. After initially converging to obtain high reward, $Q_x$ appears to get "bored" and will focus on the reward threshold, stopping short or jumping back and forth across it, which results in reduced reward but higher TD-error. This behavior is distinctive of TD-error seeking over state novelty seeking, as such states are not novel compared to moving past the threshold but do result in higher TD-error. Such behavior from $Q_x$ motivates $Q$ to explore the state space around the reward boundary. Example sequences of such behaviors are shown in Figure 4.

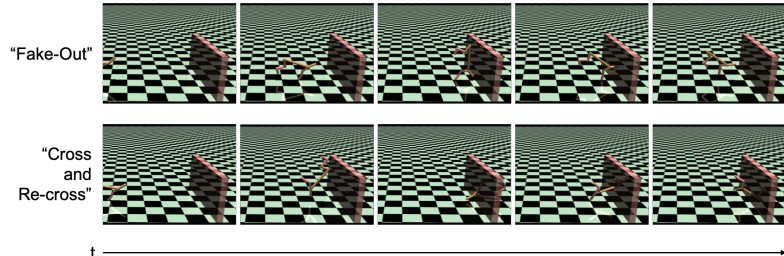

Figure 4: Example trajectories showing $Q_x$'s behavior late in training that is distinctive of TD-error maximization. The corresponding $Q$ network reliably achieves reward at this point. In "fake-out", $Q_x$ approaches the reward threshold and suddenly stops itself. In "cross and re-cross", $Q_x$ crosses the reward threshold going forward and then goes backwards through the threshold.

### 5 DISCUSSION AND CONCLUSIONS

Here, we have described a new method for using TD-error to explore in reinforcement learning. We instantiate a reward function using TD-error, and show that when combined with neural network approximation, it is sufficient to discover solutions to challenging exploration tasks in fewer training iterations than recent state novelty-based exploration methods. We hope that our results can spur further work on diverse exploration signals in RL.

It is also worth noting that there may be additional benefits provided by $Q_x$ for Q learning in non-exploration contexts. Maximizing TD-error can be seen as a form of hard example mining, and for complex tasks could result in better generalization behavior and faster transfer to new tasks through efficient trajectory sampling by $Q_x$.

One potential future area of investigation is in our method's connection to biological models of dopamine pathways in the brain where levels of dopamine correlate with TD-error in learning trials (Niv et al., 2005), a phenomenon previously described in animals (Arias-Carrión & Pöppel, 2007).

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

Table 2: Parameters used for benchmark runs.

| DEFAULT PARAMETERS | |
|---|---|
| CEM | |
| ITERATIONS | 4 |
| NUMBER OF SAMPLES | 64 |
| TOP K | 6 |
| ALL NETWORKS | |
| NEURONS PER LAYER | 256 |
| NUMBER OF LAYERS | 3 |
| NON-LINEARITIES | ReLU |
| OPTIMIZER | ADAM |
| ADAM MOMENTUM TERMS | $\beta_1 = 0.9, \beta_2 = 0.99$ |
| TRAINING | |
| Q LEARNING RATE | 0.001 |
| BATCH SIZE | 128 |
| TIME DECAY $\gamma$ | 0.99 |
| TARGET Q-FUNCTION UPDATE $\tau$ | 0.005 |
| TARGET UPDATE FREQUENCY | 2 |
| TD3 POLICY NOISE | 0.2 |
| TD3 NOISE CLIP | 0.5 |
| TRAINING STEPS PER ENV TIMESTEP | 1 |
| QXPLORE-SPECIFIC | |
| $Q_x$ LEARNING RATE | 0.001 |
| $Q$ BATCH DATA RATIO | 0.75 |
| $Q_x$ BATCH DATA RATIO | 0.75 |
| $\beta_Q$ (Q INITIAL OUTPUT BIAS) | 0 |
| RND-SPECIFIC | |
| PREDICTOR NETWORK LEARNING RATE | 0.001 |
| EXTRINSIC REWARD WEIGHT | 2 |
| INTRINSIC REWARD WEIGHT | 1 |
| $\gamma_E$ | 0.99 |
| $\gamma_I$ | 0.99 |
| DORA-SPECIFIC | |
| $\epsilon$ | 0.1 |
| $\beta$ | 0.05 |
| $\gamma_E$ | 0.99 |
| $\gamma_Q$ | 0.99 |
| $\epsilon$-GREEDY-SPECIFIC | |
| $\epsilon$ | 0.1 |

## A    IMPLEMENTATION DETAILS AND HYPERPARAMETERS

We describe here the details of our implementation and training parameters. We held these factors constant and used a shared codebase for QXplore, RND, and $\epsilon$-greedy to enable a fair comparison. We used an off-policy Q-learning method based off of TD3 (Fujimoto et al., 2018b) and CGP (Simmons-Edler et al., 2019) with twin Q-functions and a cross-entropy method policy for better hyperparameter robustness. Each network ($Q_\theta$, $Q_{x,\phi}$, RND's random and predictor networks) consisted of a 4-layer MLP of 256 neurons per hidden layer, with ReLU non-linearities. We used a batch size of 128 and learning rate of 0.001, and for QXplore sampled training batches for $Q$ and $Q_x$ of 75% self-collected data and 25% data collected by the other Q-function's policy as described in Algorithm 1.

For DORA (Fox et al., 2018), we used the hyperparameters and training procedure specified by the original paper where possible, though it was necessary to adapt the method somewhat to the continuous action domain. This is because the original formulation proscribed an "LLL" action selection scheme that requires taking discrete log-probabilities of the distribution of Q and E values over actions, which is not tractable in continuous action spaces. Instead, we tried selecting actions using either a CEM policy that maximizes the sum of the two objectives, or using the E values as a reward bonus for training Q and selecting actions that maximize Q only. We thus expect the performance of our implementations to be somewhat worse than a hypothetical distributional-DORA,

though the action selection scheme we used does make this version directly comparable to QXplore and RND. Both formulations behaved similarly on `SparseHalfCheetah` and did not achieve reward with any frequency.

For $\epsilon$-greedy sampling with continuous actions, we sampled a uniform distribution of the valid action range (-1 to 1 for all tasks) with probability $\epsilon$ and act greedily otherwise. We note that the stochastic cross-entropy method policies we used for all experiments also introduce some amount of local exploration through noisy action selection.

We present the parameters we used for the benchmark tasks in Table 2.

## B   ENVIRONMENT DETAILS

We use the `SparseHalfCheetah` environment proposed by Houthooft et al. (2016) in which a simulated cheetah receives a reward of 0 if it is at least 5 units forward from the initial position and otherwise receives a reward of -1. We also use the OpenAI gym tasks, `FetchPush`, `FetchSlide`, and `FetchPickAndPlace`, which were originally developed for benchmarking HER (Andrychow-icz et al., 2017). The objective in these environments is to move a block to a target position, with a reward function returning -1 if the block is not at the target and 0 if it is at the target. For consistency in reward shaping, we structured the reward function of the `SparseHalfCheetah` task to match the `Fetch` tasks, such that the baseline reward level is -1 while a successful state provides 0 reward, but report reward values on a 0 to 500 scale for direct comparison with previous work. We trained each method with 5 random seeds for 5,000 episodes on `SparseHalfCheetah` and 50,000 episodes on `Fetch` tasks. Time to convergence on these tasks for any exploration method is highly variable, and as such we visualize the mean and standard deviation of the runs in our results.

## C   ABLATIONS

To study the effects of different components of QXplore, we performed several ablations, as discussed in Section 4.4. First, we replaced $Q_\theta$ with simple 1-step reward prediction, and $Q_x$'s objective function with maximizing cumulative 1-step reward prediction error plus extrinsic reward, which we label as "QXplore-1-step" in Figure 5. This ablation fails to find reward, as the 1-step reward prediction error makes long range exploration more difficult to learn.

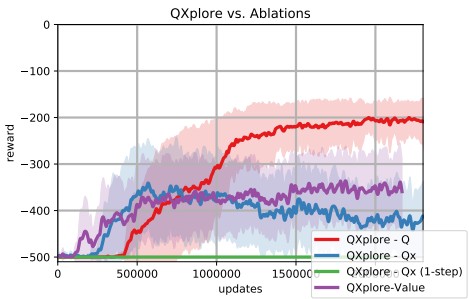

Figure 5: Plot showing the performance of two ablations, 1-Step Reward Prediction (QXplore-1-step) and Single-Policy QXplore (QXplore-value), compared to the original QXplore method. In the 1-Step ablation, $Q_x$ is trained to predict a combination of extrinsic reward and reward prediction error, and fails to make progress. In the Single-Policy ablation, the policy converges faster, but to a worse policy than vanilla QXplore due to the need to balance TD-error and extrinsic reward maximization.

Next, we tested a variant of QXplore using only a single sample policy and treating TD-error as a reward bonus, more in line with traditional exploration bonus methods. We trained a value function $V_\theta(s)$ trained via bootstrap and computed $r_x$ as $r_{x,\theta}(s_t, a_t, s_{t+1}) = |V_\theta(s_t) - (r_{\mathrm{E}}(s_t, a_t) + \gamma V'_{\theta'}(s_{t+1}))|$. This variant uses only a single sample policy, $Q_x$, which is trained via bootstrapped off-policy Q-learning using one-step reward targets $r_1 = (r_x(s_t, a_t, s_{t+1}) + \alpha r_{\mathrm{E}}(s_t, a_t))$ to maximize a combination of intrinsic and extrinsic rewards, controlled by the hyperparameter $\alpha$. We used $\alpha = 0.1$, which we found to work well for `SparseHalfCheetah` in tuning experiments. We

used a value function $V_\theta(s)$ rather than a Q-function for this ablation to avoid the wildly optimistic max action selection fully off-policy Q-functions have been reported to suffer from (Fujimoto et al., 2018a). We label this experiment as "QXplore-value" in Figure 5. This variant performs comparably to the $Q_x$ function of normal QXplore, but performance does not decrease late in training thanks to the extrinsic reward signal. However, overall performance is still well below that of normal QXplore's exploitation policy, which does not have to satisfy two conflicting training objectives.

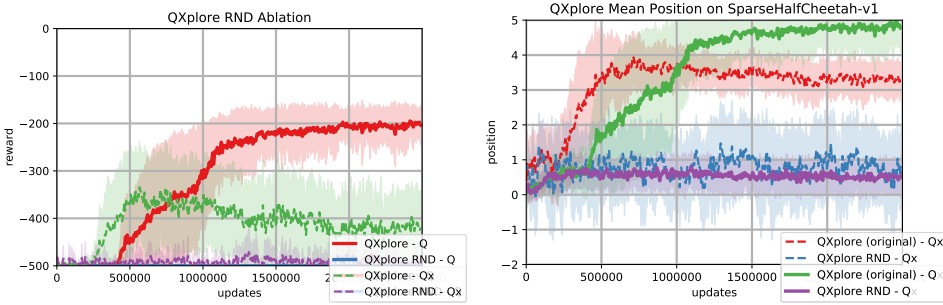

Figure 6: Plots showing the performance of QXplore where the objective of $Q_x$ is replaced by the RND exploration objective, as well as the mean position of the cheetah during an episode throughout training. While $Q_x$ does sample reward, it does so too infrequently to guide $Q$ to learn the task. While the $Q_x$ function of QXplore-RND does reach states far from the origin, it does not display directional preference, whereas original QXplore's $Q_x$ function converges to sample states around the reward threshold at 5 units.

Third, we tested a variant of QXplore where the TD-error maximization objective of $Q_x$ was replaced by the RND random network prediction error maximization objective. We call this ablation "QXplore-RND" and results are shown in Figure 6 for both $Q$ and $Q_x$ policies. We observe that neither function converges to achieve reward. While we see that $Q_x$ does sample reward, $Q$ samples reward only during two episodes of training, and $Q_x$ does not converge to achieve high expected rewards itself. Looking at the mean position of the cheetah during an episode over training, we observe that for QXplore-RND $Q_x$ samples states relatively far from the origin compared to $Q$, based on the wider standard deviation, but does not display a directional preference (besides the inbuilt tendency to move forward more readily than backward that the cheetah has built in), since states found in both directions are equally novel. Comparatively, the $Q_x$ function of normal QXplore displays a strong forward preference once reward is found, and converges on sampling states close to the 5-unit reward threshold (this results in a mean position less than 5 due to time spent traveling from the origin), while the corresponding $Q$ function prefers to move well past the reward theshold (a mean position above 5) to reliably achieve reward.

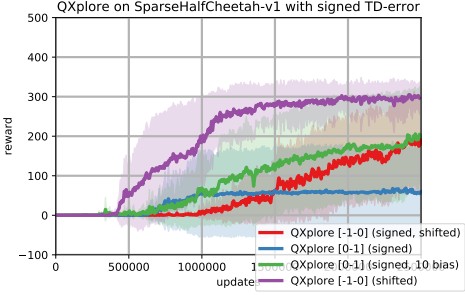

Figure 7: The performance of QXplore's $Q$ function with $Q_x$ maximizing signed versus unsigned TD-error on two different reward variants of `SparseHalfCheetah`. While $Q$ is able to learn the task for all variants, performance is reduced with the signed objective. Performance on the `SparseHalfCheetah` variant with -1 to 0 reward function is shown shifted to match the axes of the 0 to 1 variant for comparison.

Finally, we tested maximization of signed TD-error by $Q_x$ rather than unsigned. This objective tracks closer to dopamine-seeking in animals, where increases in dopamine (corresponding to an

unexpectedly positive outcome) are sought out while decreases in dopamine (from unexpectedly negative outcomes) are avoided. To emulate this, we negate the signed TD error such that negative TD-error (the predicted Q-value was less than the target value) is maximized, while positive TD-error is minimized. $Q_x$ is otherwise trained as normal. The results are shown for both variants of the `SparseHalfCheetah` reward function (-1 to 0 and 0 to 1) in Figure 7, with and without setting the initial output bias of $Q$ to 10 in the 0 to 1 case. We observe that while QXplore does train with signed TD-error, performance is reduced. While this result bares further investigation, we hypothesize this is because prior to finding reward the sign of the TD-error is not correlated with the novelty of a state, thus the state novelty exploration phase is less efficient.

## D PARAMETER SWEEPS

We performed two sets of parameter sweeps for QXplore: varying the learning rates of $Q$ and $Q_x$, and varying the ratios of data sampled by each Q-function's policy used in training batches for each method. For learning rate, we tested combinations (QLR, QxLR) (0.01, 0.01), (0.01, 0.001), (0.001, 0.01), (0.001, 0.001), (0.001, 0.0001), (0.0001, 0.001), (0.0001, 0.0001).

For batch data ratios, we tested combinations (specified as self-fraction for $Q$, then self-fraction for $Q_x$) of (0, 1), (0.25, 0.75), (0.5, 0.5), (0.75, 0.25).

Results for these sweeps can be seen in Figures 8 and 9. QXplore is sensitive to learning rate, but relatively robust to the training data mix, to the point of $Q$ training strictly off-policy with only modest performance loss.

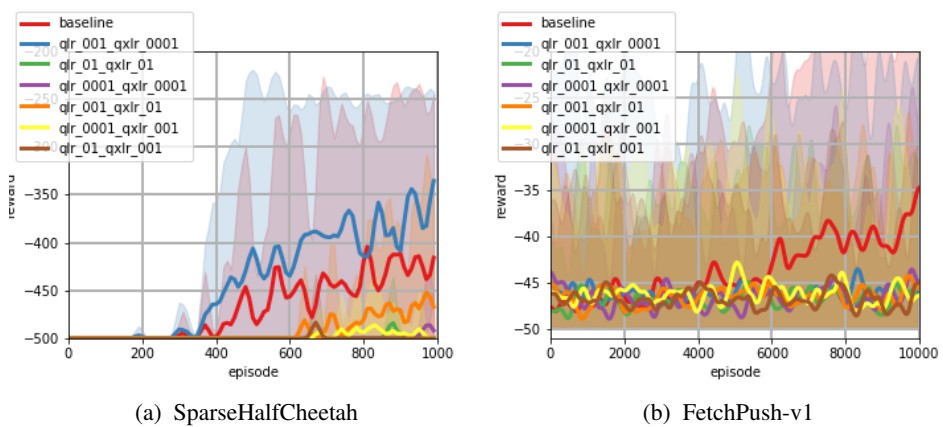

(a) SparseHalfCheetah            (b) FetchPush-v1

Figure 8: Learning rate sweeps for $Q$ and $Q_x$

### D.1 RND PARAMETER SWEEPS

As we have adapted RND to operate with vector observations and continuous actions, we performed several hyperparameter sweeps to ensure a fair comparison. We report in Figure 10 the results of varying both predictor network learning rate "lr" and extrinsic reward weight "rw" independently on the `SparseHalfCheetah` task. The baseline values for these parameters used elsewhere are 0.001 and 2 respectively. We observe that RND is fairly sensitive to reward weight, but a value of 1 or two performs well, while a learning rate of 0.001 appears to learn faster early in training without loss of final performance.

## E REWARD SHIFTING VERSUS $\beta_Q$

For most of our experiments with the `SparseHalfCheetah` environment, we used a reward function that is -1 for non-goal states and 0 for goal states to have consistent reward shaping with the `Fetch` tasks in the OpenAI Gym. However, as the original `SparseHalfCheetah` proposed by Houthooft et al. (2016) used a reward function that is 0 for non-goal states and 1 for goal states,

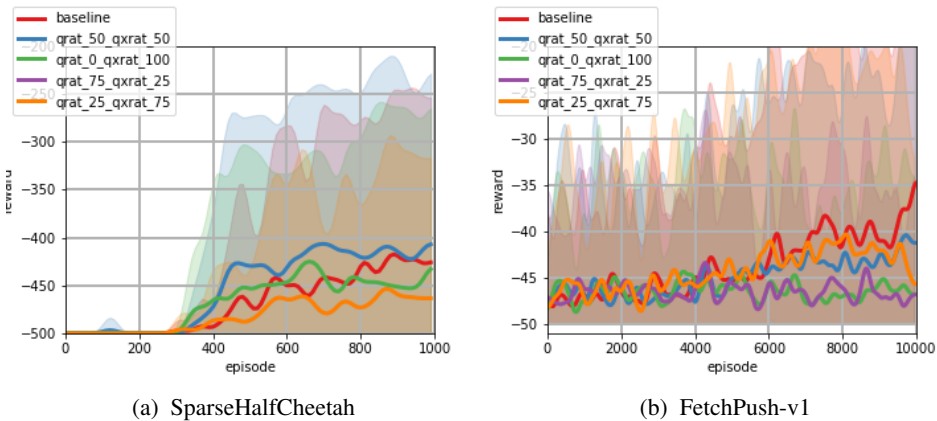

(a) SparseHalfCheetah          (b) FetchPush-v1

Figure 9: Sample ratio sweeps for $Q$ and $Q_x$

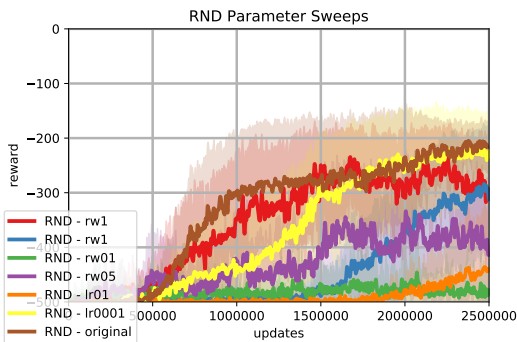

Figure 10: Parameter sweeps for RND. A reward weight of either 1 or 2 works best, with a learning rate of 0.0001 a close second.

we present results for QXplore and our implementation of RND on the original reward function as well in Figure 11. Because we initialized the output distribution $Q$ to be close to 0 initially, QXplore performed worse on this reward function due to the much smaller magnitude of TD-errors during the initial reward-free exploration phase slowing down exploration. However, adjusting the hyperparameter $\beta_Q$, the initial bias of the output neuron of $Q$, allows us to obtain identical performance to the -1 to 0 reward function. QXplore's state novelty search efficiency is sensitive to this initial TD-error magnitude, which varies depending on the reward function, but in a coarse parameter sweep of initial biases of -10, 0, 1, 10, and 100 we found a good setting for the parameter which performed comparably to the -1 to 0 reward function. This dependency is similar in concept to the reward weighting used by many reward bonus methods to trade off between exploration and exploitation, the setting of which may also depend on the reward landscape.

## F  THE 'NOISY TV' PROBLEM

The 'Noisy TV' problem is a classic issue with some state-novelty exploration methods in which states with unpredictable observations serve as maxima in the novelty reward space. QXplore's TD-error objective is not fundamentally vulnerable to the problem, but to demonstrate that our function approximation early in training is also no subject to it, we trained QXplore on a variant of the SparseHalfCheetah task where we add a random normally-distributed value to the observation vector of the agent. The variance of this noise value increases proportionately to the movement of the cheetah in the negative direction (away from the reward threshold). An agent vulnerable to the noisy tv problem will be enticed to explore in the negative direction rather than forward, as this maximizes the novelty/unpredictability of the observations.

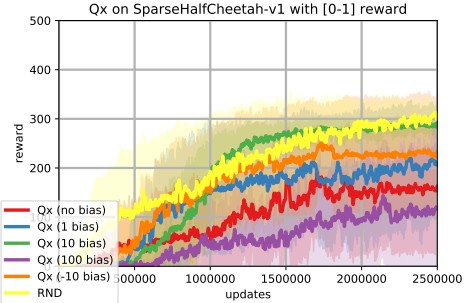

Figure 11: QXplore performance on `SparseHalfCheetah` with the 0 to 1 reward function. Adjusting $\beta_Q$ recovers full performance compared to the -1 to 0 reward function (shown here shifted by 500 reward for comparison). We tested several different values for $\beta_Q$ and found that a value of 10 worked best for `SparseHalfCheetah`.

We show the results of training QXplore on this environment in Figure 13 for both $Q$ and $Q_x$, as well as the mean position of the cheetah along the movement dimension during $Q_x$'s training rollouts. As expected, the performance of neither $Q$ nor $Q_x$ is meaningfully altered relative to the baseline, and $Q_x$ is not biased to explore backwards to a greater degree than it typically does early in training.

## G   WEIGHT INITIALIZATION

As we use neural net function approximation error as a state novelty baseline for early exploration, the behavior of $Q_x$ may be sensitive to weight initialization. To test this, in addition to the Pytorch default initialization method "Kaiming-Uniform," (He et al., 2015) which we used for all runs outside this section, we also tested initializing both $Q$ and $Q_x$ with "Kaiming-Normal" and "Xavier-Uniform," (Glorot & Bengio, 2010) two other initialization methods that result in higher variance between initial outputs of the networks, which translates into reduced training performance. We further tested two naive distributions that produced very high variance in outputs, "Normal," sampling weight values from $\mathbf{N}(0, 1)$ and "Uniform," sampling values from $\mathbf{U}(-1, 1)$. These configurations were not expected to perform as well as "Kaiming-Uniform", but do test the ability of $Q_x$ to explore given a poor initialization. In all cases other than "Kaiming-Uniform" we set the bias of each neuron to 0. The results of QXplore with each initialization scheme on `SparseHalfCheetah` are shown in Figure 12.

"Kaiming-Normal" and "Xavier-Uniform" both showed moderate decrease in overall performance, though both $Q$ and $Q_x$ were able to converge on reward. "Normal" and "Uniform" however both more-or-less prevented $Q$ from converging on reward. Their effect on the ability of $Q_x$ to find reward however is much more mild- only "Normal" and to a lesser extent "Uniform" caused significant issues with discovering and converging on reward. This suggests that $Q_x$ is not particularly dependent on careful weight initialization to explore with function approximation error.

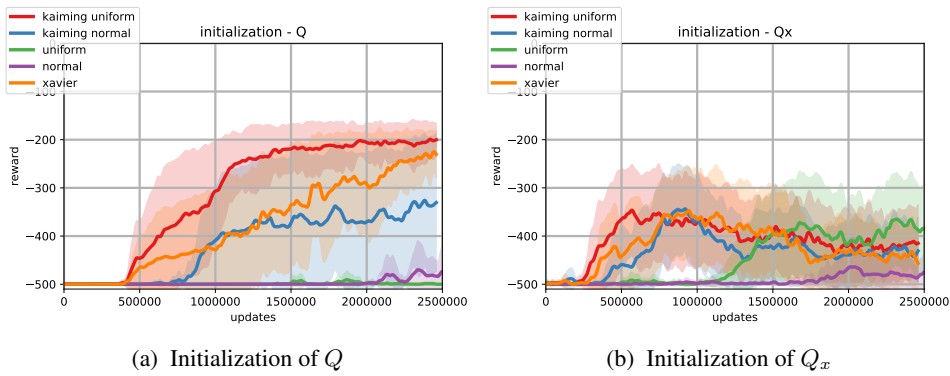

(a) Initialization of $Q$

(b) Initialization of $Q_x$

Figure 12: Several alternate initialization schemes for $Q$ and $Q_x$. While $Q$ is adversely impacted, $Q_x$ is relatively robust even to very poor initializations such as "Normal" and "Uniform."

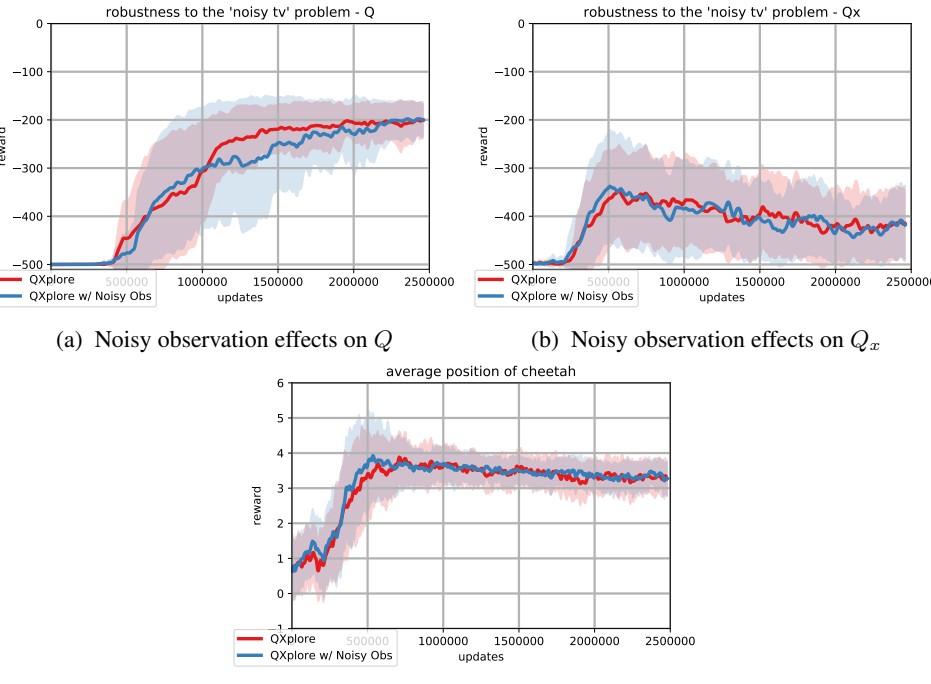

(a) Noisy observation effects on $Q$

(b) Noisy observation effects on $Q_x$

(c) Noisy observation effects on absolute position

Figure 13: QXplore trained on a 'noisy tv' variant of `SparseHalfCheetah` where one element of the observation vector is normally distributed random value whose variance increases if the cheetah moves in the negative direction. The performance of QXplore is not impacted in any way by this noise, and it trains as normal.

