# OpenReview forum: "QXplore: Q-Learning Exploration by Maximizing Temporal Difference Error"
_ICLR.cc/2020/Conference — Reject_

### Official Review · AnonReviewer2 · 2019-10-15
**Official Blind Review #2**

**Rating:** 3

**Review:**

The paper proposes an exploration method based on the TD-error as an adversarial reward signal. The authors show that this reward signal has interesting exploration properties. They compare it empirically to RND and epsilon-greedy, showing better performance. Besides, they perform additional ablation studies to better investigate the properties of their approach.

Though the paper proposes an interesting concept, it suffers from many weaknesses, some easy to fix and some which will require more work.

Here are some remarks, in random order:

- at the end of the introduction, the authors mention inspiration from computational neuroscience models, but they do not come back to this aspect in their work. To me, these remarks should be removed from the paper and kept for another paper about the biological significance of the model. In the conclusion, the authors come back to the "biological concepts of curiosity boredom, and exploration", I would rather say they are psychological concept, and the authors should have a look at developmental psychology and developmental robotics if they really want to contribute in this respect (but not for this paper).

- some related work references are dispersed in the introduction, in Section 2, in the beginning of Section 3.2, in the end of Section 3.3 and in a few other places. The authors should build a proper "related work" section. Globally, the paper is poorly organized, e.g. Section 3.2 refers to Section 3.4 etc.

- given that Q_x's reward function is the unsigned TD-error, I would like to see whether QXplore can deal with problems taking both some positive and negative rewards.

- the authors mention RND and DORA as baselines, but only compare to RND. What about DORA? Despite the excitement it generated when published, I suspect RND is a rather weak baseline. There are many other exploration frameworks, the paper would be much stronger if the comparison was with respect to many other methods, such as GEP-PG (Colas et al. , ICML 18), Novelty search approaches, etc. To me, the weak comparison is the biggest weakness of this work.

- the authors compared themselves to approaches based on TRPO while they were using TD3 (this information is hidden in Appendix A and should be moved in the main paper. But then, is the difference in performance due to using TRPO which is known to be less sample efficient? Again, this makes the comparison very weak, the authors should rely on the same algorithm from both sides.

- the caption of Table 1 should be explicit about which algorithm comes from which paper.

- I found Section 4.5 very weak, it looks like mere "handwaving" and would deserve a proper quantitative study if the authors want to keep it. In my opinion, the authors should remove it for now and move Appendices E and F to the main paper instead. As is, Appendix E is very poor (by the way, the caption and the figure legend do not match, so we don't know which is which) and I had to look for the number of seeds until I found it hidden in Appendix F.

- Appendix C shows that, though the authors try to minimize the importance of this fact, their algorithm is very sensitive to initialization. It is very honest of them to have kept this study in their paper, but I'm afraid it strongly speaks against the algorithm.

- As described, SparseHaflCheetah is not that sparse (-1 everywhere and 0 when you succeed, as this reward scheme already favors exploration). It would be more informative to use 0 everywhere and 1 when successful.

typos:

Eq (1) and (3) should finish with a dot as it is the end of a sentence.
p3: MDP's => MDPs
Section 3.3, second sentence: avoid starting a sentence with a symbol.
Section 4, the second sentence is not a sentence (no main verb)
p14: is also no subject to it => not

**Experience Assessment:**

I have read many papers in this area.

**Review Assessment: Checking Correctness Of Derivations And Theory:**

N/A

**Review Assessment: Checking Correctness Of Experiments:**

I assessed the sensibility of the experiments.

**Review Assessment: Thoroughness In Paper Reading:**

I read the paper at least twice and used my best judgement in assessing the paper.

---

> ### Author Response · Authors · 2019-11-13
> **Responses to Points Raised**
>
> Thank you for your helpful comments on organization and clarity, and for taking the time to collect typos. We have responded to several of the points raised below:
>
> - “inspiration from computational neuroscience models...should be removed from the paper and kept for another paper”
>
> Echoing our response to Reviewer 1, this is a reasonable criticism. We leave this for future work, and have revised our manuscript accordingly.
>
> - “some related work references are dispersed…[and] the paper is poorly organized”
>
> We have condensed some of these sections, and reworked some of the confusing organization in the updated manuscript. We will further aggregate related work into a discrete section in a future draft.
>
> - “...whether QXplore can deal with problems taking both some positive and negative rewards”
>
> Informally, we have tested QXplore on tasks with both positive and negative rewards, and found performance to be unaffected, as TD-error is relative to the current predicted Q-value rather than the absolute reward scale.
>
> - “the authors mention RND and DORA as baselines, but only compare to RND. What about DORA?”
>
> We decided that DORA would not be a useful comparison, because it has not been demonstrated to be effective on large MDPs, and isn’t directly applicable to continuous action-space problems. We opted to focus on RND because of its algorithmic relationship to QXplore and because it has been shown to work on large MDPs.
>
> - “the paper would be much stronger if the comparison was with respect to many other methods...”
>
> This is an understandable position. However, it is complicated by the fact that in its current form QXplore is formulated to work only  with an off-policy RL algorithm, and nearly all other existing exploration methods (including RND, VIME, EMI, ICM, etc.) are implemented on top of policy gradient methods such as PPO and TRPO. We attempted to adapt several of these methods for use in off-policy learning settings, but found we couldn’t get good performance on a Q-learning baseline using author-provided hyperparameters for any method except for RND. GEP-PG is a reasonable method to compare with, and we will attempt to make a fair comparison by the end of the rebuttal period.
>
> - “is the difference in performance due to using TRPO which is known to be less sample efficient?”
>
> We recognize that some of our performance gains reported in Table 1 may be attributed to this difference. However, the maximum performance achieved by QXplore is significantly higher than the maximum performance achieved by other methods in any amount of time as reported by the original authors.
>
> - “the caption of Table 1 should be explicit about which algorithm comes from which paper”
>
> Good suggestion. We’ve updated the manuscript accordingly.
>
> - “...their algorithm is very sensitive to initialization.”
>
> The alternate initialization schemes tested were intended to test the performance of QXplore on suboptimal initializations, not to provide equivalent initialization schemes to Kaiming-Uniform. An informal, empirical spot check reveals a relationship between the performance of agents using a given initialization method and the mean values output by the initialized network. Concretely, when 128 experiences are sampled randomly in the SparseHalfCheetah environment, the standard Kaiming-Uniform initialization produces mean Q-values in the [-0.15, 0.15] range, Xavier-Uniform in the [-1.5, 1.5] range, Kaiming-Normal in the [-5, 5] range, Uniform in the [-200, 200] range, and Normal in the [-2000, 2000] range. These increases in initial Q-value range correlate with the decreases in performance when we train QXplore using these initializations (see new Figure 11 in Appendix G). Furthermore, a relationship between training speed/final performance and feature/output magnitudes has been repeatedly shown in the literature (notably in Batch Norm [1]). Our initialization experiments were intended to demonstrate robustness to suboptimal initializations, and show that Qx is still able to find reward given even very bad initializations.
>
> - “SparseHalfCheetah is not that sparse... It would be more informative to use 0 everywhere and 1 when successful”
>
> To allay any concerns about the change in reward function, we ran QXplore on the 0 to 1 reward function and include results in a revision to our paper. Briefly, without modification QXplore trains slower on this reward function, but adding a hyperparameter to initialize the bias of the output neuron of Q to a nonzero value can correct this slowdown. Among a simple parameter sweep with values of 1, 10, and 100, we found that an initial bias of 10 recovers the full performance we reported with the -1 to 0 reward function. Full details can be found in Appendix E.
>
> - “typos”
>
> Good catch - thanks. We’ve fixed them in the updated manuscript.
>
> [1] Ioffe, S., & Szegedy, C. (2015). Batch normalization: Accelerating deep network training by reducing internal covariate shift. arXiv preprint arXiv:1502.03167.

---

### Official Review · AnonReviewer3 · 2019-10-23
**Official Blind Review #3**

**Rating:** 3

**Review:**

This paper proposes an exploration algorithm for reinforcement learning agents based on learning a separate Q-value function which treats TD-errors as rewards. Experiments on continuous control tasks with a difficult exploration component are used to highlight the effectiveness of the approach.

The proposed algorithm is interesting and has an intuitive appeal, the magnitude of the TD-error as an "auxiliary reward" seems like a natural choice to guide exploration. The experiments are comprehensive, including some ablation studies and robustness tests, with one caveat. Due to some experimental details (particularly concerning the environments), the empirical results may be invalid and it is difficult to assess them. There are also some certain parts that are unclear in the experimental results. Overall, because of these issues, I cannot recommend acceptance although I would be willing to increase my score if my concerns are addressed.

Concerning the SparseHalfCheetah task, from Appendix F: "...baseline reward level is -1 while a successful state provides 0 reward, but report reward values on a 0 to 500 scale for direct comparison with previous work". This reward function is not sparse since it gives -1 at every step except at the goal which would incentivize the agent to explore naturally if it were acting greedily (or approximately greedily) with respect to the value function. For a sparse reward task, I would expect the reward to be 0 everywhere except at the goal state(s), where it would have some positive value. This is the case for the original reward function for SparseHalfCheetah described in [1]. This change to the reward function makes any comparisons to previous work questionable (Table 1) and could also affect the qualitative results in section 4.5.

Similarly, the other 3 tasks also do not have a sparse reward structure as the agent receives -1 at every timestep. This again is a confounder for assessing the exploration capabilities of the algorithm. It would be preferable to use other environments such as those in [1], which indeed have sparse rewards.

For these sparse reward problems where most rewards are zero, then before any nonzero reward is observed, it seems like Qx would mostly behave like previous algorithms based on a state novelty term (as discussed in sec 3.4). This could happen since the agent would only be observing rewards of 0 at this point during training. It is unclear to me then which benefits QXplore could have compared to previous algorithms such as DORA.

Other points:
1) Some discussion of possible pitfalls for the algorithm could be added. For example, while the noisy TV problem may be avoided, there could be a noisy reward problem now. The absolute TD error could be high simply due to stochasticity in rewards. The agent could also be vulnerable to scenarios in which a stochastic transition brings the agent to either a state eventually leading to very high reward or to a state leading to no reward as this would cause the TD error be high (even if the value function converges).
2) The sentence before sec 3.3, "Further TD-error-based exploration with a dedicated exploration policy removes the exploitation-versus-exploration
tradeoff that ... to an optimal Q-function." Could the authors clarify how TD-error-based exploration avoids the exploitation-versus-exploration
tradeoff? I do not see the connection here.
3) In the experiments, how are actions chosen for the epsilon-greedy baseline since the action space is continuous (and not discrete)?
4) I am bit confused as to what the difference between the lines Q and Qx are in Fig. 3.
5) Why was DORA not included as a baseline algorithm? It would seem to be the most closely related to QXplore when rewards are mostly constant (or zero).
6) Are two separate buffers necessary instead of a single shared buffer (with uniform sampling)? It seems to needlessly introduce complexity and additional hyperparameters (the ratios of samples from each buffer). If this is a key choice, then it should be mentioned in the paper.
The text could also be clarified to indicate there are two buffers. In some places the writing suggests there is only one, e.g. sec3.3 "...policy with a replay buffer shared between Q_\theta andthe Q-function maximizing rx, which we term Qx."
7) In section 4.4, for the "Single-Policy QXplore", why was Q(s,a) replaced by V(s)? If I understand correctly, this experiment tries to test a variant where the TD error is used as a reward bonus. If this is the case, it seems like the best comparison would be to leave the original Q(s,a) which is learned by Q-learning instead of changing it to a state-value function.

Suggestions (did not impact score)
- I wonder if there is a connection between Qx and the variance of the returns since the latter can be learned by using the squared TD-errror as a reward (see the Direct Variance algorithm in [2]) and the absolute TD-error is a similar quantity. In this way, it could be possible to frame QXplore as following a type of risk-seeking stategy (see [3] for an algorithm that makes use of the variance).
- I am not sure 'adversarial' is the right term to describe Q and Qx since the two policies are not in direct competition with each other.

[1] "VIME: Variational Information Maximizing Exploration" by Houthooft et al.
[2] "Directly Estimating the Variance of the λ-Return Using Temporal-Difference Methods" by Sherstan et al.
[3] "Deep Reinforcement Learning with Risk-Seeking Exploration" by Dilokthanakul and Shanahan


**Experience Assessment:**

I have published one or two papers in this area.

**Review Assessment: Checking Correctness Of Derivations And Theory:**

N/A

**Review Assessment: Checking Correctness Of Experiments:**

I carefully checked the experiments.

**Review Assessment: Thoroughness In Paper Reading:**

I read the paper at least twice and used my best judgement in assessing the paper.

---

> ### Author Response · Authors · 2019-11-13
> **Responses to Points Raised**
>
> Thank you for your helpful and supportive comments. We are pleased you found our method interesting and intuitive. We have addressed several points below, and hope that we have addressed your concerns sufficiently:
>
> - “[SparseHalfCheetah’s] reward function is not sparse since it gives -1 at every step except at the goal”
>
> Our use of the term “sparse” relates to the fact that the reward function is a step function (-1 until the cheetah passes a threshold, 0 afterwards), where the 0-valued region is unlikely to be sampled by a naive exploration policy. Since there are two discontinuous, uniform regions in the reward space, the boundary between which is distant from the starting point, we still consider this to be a “sparse” reward function. Naive exploration algorithms such as epsilon greedy are not capable of exploring it (see Figure 3).
>
> - “This [default -1] change to the reward function makes any comparisons to previous work questionable”
>
> To allay any concerns about the change in reward function, we ran QXplore on the 0 to 1 reward function and include results in a revision to our paper. Briefly, without modification QXplore trains slower on this reward function, but adding a hyperparameter to initialize the bias of the output neuron of Q to a nonzero value can correct this slowdown. In a small hyperparameter sweep we found that an initial bias of 10 recovers the full performance we reported with the -1 to 0 reward function. Full details can be found in Appendix E.
>
> - “It is unclear to me then which benefits QXplore could have compared to previous algorithms such as DORA.”
>
> QXplore will not outperform previous state novelty methods such as RND or DORA before any reward is found. However, once reward is encountered, Qx will focus exploration on states that lead to unexpected reward (or an unexpected absence of reward), and as such this focused exploration results in faster learning and convergence by Q.
>
> - “Some discussion of possible pitfalls for the algorithm could be added”
>
> You are correct in your assessment that stochastic rewards create local (but not necessarily global) maxima in the unsigned TD-error landscape, which can cause QXplore to explore insufficiently. We observe that one simple solution is to use signed TD-error instead of unsigned, which avoids this issue at the cost of worse performance on SparseHalfCheetah (based on new results in Appendix E).
>
> - “Could the authors clarify how TD-error-based exploration avoids the exploitation-versus-exploration tradeoff?”
>
> The intent of this passage was to highlight the fact that driving exploration using TD-error will inherently sample more states that are relevant to exploitation, whereas state-novelty exploration methods are reward agnostic and tend not to sample as many exploitation-relevant states in sparse environments. We have revised the manuscript to make this distinction clearer.
>
> Additionally, we have revised the manuscript to include an experiment that demonstrates how a state-based exploration objective (here, RND) instead of the TD-error-based objective for the Qx policy is insufficient to train Q, and demonstrate that neither policy converges to reward.
>
> - “how are actions chosen for the epsilon-greedy baseline since the action space is continuous (and not discrete)?”
>
> With probability 0.1 we sampled from a uniform distribution between the minimum and maximum action values (typically -1 and 1).
>
> - “what is the difference between the lines Q and Qx are in Fig. 3.”
>
> In Figure 3, Q is the performance of our exploitation policy implied by Q (which maximizes reward), while Qx is the performance of our exploration policy implied by Qx (which maximizes the TD-error of Q). We have relabled this figure in our revised draft to avoid confusion.
>
> - “Why was DORA not included as a baseline algorithm?”
>
> DORA was not included as a baseline because DORA was never tested on large MDPs, and thus we were unsure how well it would perform by itself on such cases. We focused on RND instead as it is also related to the behavior of QXplore when no reward has been found, but has been demonstrated on large MDPs previously.
>
> - “Are two separate buffers necessary instead of a single shared buffer (with uniform sampling)?”
>
> We have revised the paper to clarify that two buffers were used. While not essential, we found that using a skewed sampling ratio of self versus non-self data results in higher performance, possibly by reduced non-self data decreasing off-policy variance/stability issues. See Appendix D for empirical results.
>
> - “why was Q(s,a) replaced by V(s)?”
>
> We opted for V(s) over Q(s,a) for the single-policy variant due to concerns about inaccurate target Q value prediction when training Q fully off-policy, and also to demonstrate that the QXplore paradigm works for value functions as well as Q-functions. We expect this variant to outperform the simpler ablation, and thus represent a better version of 1-policy QXplore.

---

### Official Review · AnonReviewer1 · 2019-10-24
**Official Blind Review #1**

**Rating:** 3

**Review:**

The authors present a novel exploration method wherein an additional Q-function/policy is learned that treats abs(TD-error) of the standard Q-function as its reward function. Both policies are executed in parallel and experience is shared between them for off-policy learning. They demonstrate their method's superiority on a few continuous control tasks relative to RND. Overall, I thought the method was interesting and novel, but several concerns prevent me from endorsing its publication.

1) Incorrect or untuned RND baseline.
As the tasks under consideration are quite different from those in the RND paper, RND must be adapted significantly. It is therefor troubling that only a single RND-specific hyper-parameter is reported, when there should be many (e.g. ratio of intrinsic to extrinsic rewards, see Table 5 in their paper for many more).

It could be that you're adapting RND to be more like an ablation of your own approach, such as replacing the Qx TD-error rewards with random network distillation rewards. This is fine (and I'd suggest doing this as an additional baseline if not), but it should be labeled as such.

2) No comparisons to published RND baselines.
All of these problems could be avoided if the authors chose to run their method on tasks with published baseline results. Indeed, the lack of a Montezuma's Revenge run is particularly glaring. The author's are right to point out that Atari is perhaps not ideal for exploration since most novel states are rewarding, so I'm not expecting it to outperform RND per se. But the authors also claim that in these situations it won't do worse than RND, and without a Montezuma's Revenge run, this claim isn't well founded empirically.

3)No comparison to value uncertainty methods.
This is also made more frustrating when contrasted to the author's claim that "to our knowledge posterior uncertainty methods have thus-far only been demonstrated in small MDP." Osband, Aslanides, and Cassirer (2018) has results on a continuous control task and Montezuma's Revenge. As QXplore is related to both RND and this class of value uncertainty methods, is feels as though the latter should also be included.

Smaller points:

* No mention of prioritized experience replay. This method utilizes TD-error to bias replay rather than act explicitly, but there seems to be an interesting connection (acting to obtain more TD-error vs over-sampling high TD-error transitions) that was disappointingly ignored in this work.

* I'd suggest dropping the "biologically inspired" bits from the paper. The evidence you're citing (dopamine represents TD error + dopamine seeking behavior) could be used to justify any approach that optimistic with respect to value function uncertainty (e.g. thompson sampling on value function posterior), and doesn't really add anything.

* While the Fetch results are impressive, they should be contextualized by the results obtained in the HER paper. Obviously, HER isn't as general as this method, but it is worth reminding the reader that there is still a large performance gap between QXplore and the SOTA on these tasks.

* noisy TV problem also isn't a problem for RND. Either compare to prediction-error based approaches and show that you're more robust to this issue, or drop it entirely.

Rebuttal EDIT:

I appreciate the authors' responsiveness to my feedback on the text; the framing of biological plausibility and the connection to HER performance are both handled quite well now.

I also appreciate the efforts the authors have demonstrated in adding new baseline results. That said, GEP and DORA results seem preliminary still (which is understandable given the timeframe), and would benefit from tuning and immediate results reporting in the case of GEP.

I still find the treatment of RND problematic. "Where possible, we used the parameters specified by RND in experiments (i.e. fixing intrinsic/extrinsic reward weight at 2)" -- this is not the right attitude when testing such different domains and reinforcement learning algorithms. All of these hyper-parameters should be swept over in a manner analogous to your own approach before this comparison is valid.

I acknowledge the Montezuma's Revenge (MR) isn't representative of exploration problems by itself -- this could be a good place to raise the bar and test against all 6 or so exploration-heavy problems in the Atari Suite. But even just running on MR would be better than introducing new tasks without published results from prominent exploration methods like RND. At the very least adding MR to your set of tasks would allow for comparisons against prior exploration results, which is the primary thing preventing me from raising my score.

**Experience Assessment:**

I have published in this field for several years.

**Review Assessment: Checking Correctness Of Derivations And Theory:**

N/A

**Review Assessment: Checking Correctness Of Experiments:**

I carefully checked the experiments.

**Review Assessment: Thoroughness In Paper Reading:**

I read the paper thoroughly.

---

> ### Author Response · Authors · 2019-11-13
> **Responses to Points Raised**
>
> Thank you for your helpful comments and advice. We have address several points raised below:
>
> - “Incorrect or untuned RND baseline.”
>
> Because we had to adapt RND to the off-policy Q-learning paradigm (a change that precludes the direct use of many parameters reported in RND’s Table 5), it is reasonable to be concerned about the quality of the implementation. However, we feel that our implementation and tuning is representative of a correct adaptation. Where possible, we used the parameters specified by RND in experiments (i.e. fixing intrinsic/extrinsic reward weight at 2), and we performed several baseline experiments across multiple exploration and non-exploration tasks (although not on Atari games) that showed evidence of performance differences in line with the results reported in the original paper. To further assuage concerns about parameter tuning, we will provide parameter sweeps over learning rate and reward weighting in a revision before the end of the rebuttal period.
>
> - “...adapting RND to be more like an ablation of your own approach, such as replacing the Qx TD-error rewards with random network distillation rewards...”
>
> This is a good idea - thanks! At your suggestion, we’ve performed a new ablation where we replaced the Qx agent’s TD-error objective with RND’s state novelty objective. We found that this variant does not converge to achieve reward in the time allowed, and samples rewarding states too infrequently to train the Q function. We’ve included this experiment in our revised manuscript.
>
> - “No comparisons to published RND baselines… [including] Montezuma’s Revenge”
>
> We considered this comparison, but felt that adapting the QXplore method to the Atari domain (with its image observations and observations) would require too much of a change in the method and require correct implementation of too many non-exploration-relevant details to draw useful inference for the task on continuous control domains. Furthermore, a recent paper [1] performed rigorous comparisons of exploration methods on Atari games and found that performance on Montezuma’s Revenge is not predictive of performance more broadly within the benchmark, and other exploration methods don’t significantly outperform epsilon-greedy on Atari tasks.
>
> - “No comparison to value uncertainty methods...”
>
> Thank you for letting us know about this work! We will compare with any value uncertainty methods which can be easily evaluated on SparseHalfCheetah and add such comparisons before the rebuttal period ends.
>
> - “...dropping the ‘biologically inspired’ bits…”
>
> While we believe that the biological relevance of this method should be explored more, we will leave that for future work. We have revised the manuscript and now only mention the connection in passing.
>
> - “...Fetch Results… should be contextualized by the results obtained in the HER paper.”
>
> Good suggestion. We’ve added some description to note that QXplore is not state of the art compared to goal-directed RL methods like HER.
>
> - “Noisy TV problem also isn’t a problem for RND. Either compare to prediction-error based approaches and show that you're more robust to this issue, or drop it entirely”
>
> Our mention of QXplore’s robustness to the Noisy TV problem was intended to compare it against naive state-prediction-error approaches, not RND. We’ve revised the text with more context.
>
> [1] On Bonus Based Exploration Methods in the Arcade Learning Environment, Anonymously submitted to ICLR 2019, https://openreview.net/forum?id=BJewlyStDr

---

### Author Response · Authors · 2019-11-13
**Paper Revisions**

We thank our reviewers for their helpful comments, and for their positive assessment of the significance of our method, which represents a novel approach to the exploration problem in RL in the realistic case where reward feedback is sparse but not singular, and which performs better than state novelty exploration in cases where novel states are poorly correlated with improving reward.

We have made the following major changes to our manuscript:
1. We included results using the original SparseHalfCheetah task with rewards {0, 1}.
2. We included an ablation comparing QXplore with a similar 2-policy algorithm where the exploration policy’s objective is changed to use RND’s state-novelty objective.
3. We included an ablation comparing QXplore with a variant using signed TD-error instead of unsigned TD-error.
4. We revised several sections to clarify points of confusion.

---

> ### Author Response · Authors · 2019-11-15
> **Additional Paper Revisions**
>
> We thank the reviewers again for their thoughtful comments. We have further revised our manuscript in the last several days to include significant additional experiments requested by reviewers:
>
> - We have added two new method comparisons, to DORA and GEP-PG on SparseHalfCheetah, showing that QXplore performs favorably to both.
>
> - We have performed more hyperparameter sweeps on our RND implementation, showing that our original parametrization is comparable or better than other configurations tested.
>
> - We added some additional comparison to RND for the 0-to-1 reward function as well as QXplore. RND performs slightly better than on the -1-to-0 reward function and is on-par with QXplore for that variant, rather than slightly worse.
>
> - We revised and centralized our related work at the request of reviewer 2.
>
> - We have clarified the particularities of adapting some related methods, such as DORA and GEP-PG, to our experimental settings.
>
> We hope that these changes and additional experiments have addressed the concerns raised by our reviewers, and that the significance of our contribution to exploration has been more clearly demonstrated.

---

### Decision · Program_Chairs · 2019-12-19

**Decision:**

Reject

**Comment:**

There is insufficient support to recommend accepting this paper.  Although the authors provided detailed responses, none of the reviewers changed their recommendation from reject.  One of the main criticisms, even after revision, concerned the quality of the experimental evaluation.  The reviewers criticized the lack of important baselines, and remained unsure about adequate hyperparameter tuning in the revision.  The technical exposition lacked a sober discussion of limitations.  The paper would be greatly strengthened by the addition of a theoretical justification of the proposed approach.  In the end, the submitted reviews should be able to help the authors strengthen this paper.